# Overcoming Chemoresistance in Cancer: The Promise of Crizotinib

**DOI:** 10.3390/cancers16132479

**Published:** 2024-07-07

**Authors:** Sanaa Musa, Noor Amara, Adan Selawi, Junbiao Wang, Cristina Marchini, Abed Agbarya, Jamal Mahajna

**Affiliations:** 1Department of Nutrition and Natural Products, Migal—Galilee Research Institute, Kiryat Shmona 11016, Israel; 2Department of Biotechnology, Tel-Hai College, Kiryat Shmona 11016, Israel; 3School of Biosciences and Veterinary Medicine, University of Camerino, 62032 Camerino, Italy; 4Oncology Department, Bnai Zion MC, Haifa 31048, Israel

**Keywords:** cancer chemoresistance, crizotinib, combination therapy, tyrosine kinase inhibitor

## Abstract

**Simple Summary:**

Cancer cells often become resistant to treatment, making it harder to fight the disease effectively. This resistance can happen due to genetic changes, proteins that pump out drugs, or the environment around the tumor. Crizotinib is a drug that helps overcome this resistance in certain lung cancers and lymphomas by targeting specific proteins. It can also break down a harmful protein found in some cancers. Researchers have found that combining crizotinib with other drugs can improve its effectiveness, especially for cancers with specific genetic mutations.

**Abstract:**

Chemoresistance is a major obstacle in cancer treatment, often leading to disease progression and poor outcomes. It arises through various mechanisms such as genetic mutations, drug efflux pumps, enhanced DNA repair, and changes in the tumor microenvironment. These processes allow cancer cells to survive despite chemotherapy, underscoring the need for new strategies to overcome resistance and improve treatment efficacy. Crizotinib, a first-generation multi-target kinase inhibitor, is approved by the FDA for the treatment of ALK-positive or ROS1-positive non-small cell lung cancer (NSCLC), refractory inflammatory (ALK)-positive myofibroblastic tumors (IMTs) and relapsed/refractory ALK-positive anaplastic large cell lymphoma (ALCL). Crizotinib exists in two enantiomeric forms: (R)-crizotinib and its mirror image, (S)-crizotinib. It is assumed that the R-isomer is responsible for the carrying out various processes reviewed here The S-isomer, on the other hand, shows a strong inhibition of MTH1, an enzyme important for DNA repair mechanisms. Studies have shown that crizotinib is an effective multi-kinase inhibitor targeting various kinases such as c-Met, native/T315I Bcr/Abl, and JAK2. Its mechanism of action involves the competitive inhibition of ATP binding and allosteric inhibition, particularly at Bcr/Abl. Crizotinib showed synergistic effects when combined with the poly ADP ribose polymerase inhibitor (PARP), especially in ovarian cancer harboring BRCA gene mutations. In addition, crizotinib targets a critical vulnerability in many p53-mutated cancers. Unlike its wild-type counterpart, the p53 mutant promotes cancer cell survival. Crizotinib can cause the degradation of the p53 mutant, sensitizing these cancer cells to DNA-damaging substances and triggering apoptosis. Interestingly, other reports demonstrated that crizotinib exhibits anti-bacterial activity, targeting Gram-positive bacteria. Also, it is active against drug-resistant strains. In summary, crizotinib exerts anti-tumor effects through several mechanisms, including the inhibition of kinases and the restoration of drug sensitivity. The potential of crizotinib in combination therapies is emphasized, particularly in cancers with a high prevalence of the p53 mutant, such as triple-negative breast cancer (TNBC) and high-grade serous ovarian cancer (HGSOC).

## 1. Introduction

Crizotinib is a novel multi-target kinase inhibitor that has established itself as a valuable clinical tool in oncology. Approved by the FDA, crizotinib specifically targets anaplastic lymphoma kinase (ALK) and ROS1 (c-ros oncogene 1) mutations. This drug demonstrates efficacy in treating ALK-positive and ROS1-positive non-small cell lung cancer (NSCLC) [1,2,3]. Crizotinib extends it therapeutic applications beyond lung cancer, for example, in treating unresectable, recurrent, or refractory inflammatory ALK-positive myofibroblastic tumors (IMTs) in both adult and pediatric populations (age 1 year and older) [4]. Moreover, crizotinib was also approved for relapsed/refractory ALK-positive anaplastic large cell lymphoma (ALCL) [5].

Beyond its established targets, crizotinib exhibits inhibitory effects on the c-Met/hepatocyte growth factor receptor (HGFR) and other kinases [3]. Further research has unveiled crizotinib’s potential to synergize with standard cancer therapies, potentially overcoming chemoresistance, a significant hurdle in cancer treatment [6]. This review delves into the diverse activities of crizotinib, exploring its promising role in combating chemoresistance and its potential to expand cancer treatment options.

## 2. Crizotinib Stereoisomers

### 2.1. Enantioselective Anticancer Effects of (S)-Crizotinib

Crizotinib, a small molecule with the chemical formula 3-[1-(2,6-dichloro-3-fluorophenyl)ethoxy]-5-[1-(piperidin-4-yl)-1H-pyrazol-4-yl]pyridine-2-amine, exists as two enantiomers: (R)-crizotinib and (S)-crizotinib (Figure 1). These mirror image forms exhibit distinct biological activities due to their differing stereogenic centers. (R)-crizotinib is clinically used to inhibit ALK kinase in specific cancers [7] and is our focus in the current review. However, growing interest surrounds (S)-crizotinib for its potent anticancer properties, which are attributed to its selective inhibition of MutT Homolog 1 (MTH1) [8].

MTH1 plays a vital role in maintaining genomic stability by eliminating oxidized nucleotides (e.g., 8-oxo-dGTP, 2-OH-dATP) from the nucleotide pool, preventing their incorporation into DNA during replication. Cancer cells are often under increased oxidative stress. They exhibit a heightened dependence on MTH1 for proper DNA replication. This dependence makes MTH1 a target for cancer therapy. (S)-Crizotinib has emerged as a promising MTH1 inhibitor, demonstrating potent anticancer effects in various studies [8,9,10].

Huber et al. (2014) elucidated the enantioselectivity of MTH1 inhibition through co-crystal structures. Their findings suggest that the unfavorable binding of (R)-crizotinib compared to (S)-crizotinib is due to a conformational clash, not to specific protein interactions [11].

### 2.2. (S)-Crizotinib Demonstrates Superior MTH1 Inhibition and Potency

(S)-crizotinib exhibits remarkable selectivity for MTH1, displaying potent inhibitory activity at nanomolar concentrations, significantly exceeding its (R)-enantiomer [8,12]. This inhibition leads to the accumulation of oxidized nucleotides in the DNA, ultimately resulting in DNA damage and triggering apoptosis (programmed cell death). The IC50 value for (S)-crizotinib’s inhibition of MTH1 is considerably lower than that of (R)-crizotinib. This further highlights (S)-crizotinib’s superior potency and drug efficacy against MTH1.

### 2.3. (S)-Crizotinib Induces Apoptosis through Multiple Pathways

(S)-crizotinib treatment independently induces the generation of reactive oxygen species (ROS) within cancer cells, leading to a state of oxidative stress and ultimately apoptosis. Elevated ROS levels contribute to cellular damage and activate the endoplasmic reticulum (ER) stress pathway, further promoting cell death. This ROS production occurs independently of MTH1 inhibition, suggesting that (S)-crizotinib exerts its anticancer effects through multiple complementary mechanisms [9,11].

### 2.4. Structural and Computational Evidence for Enhanced Binding

Structural studies, including co-crystal structures, reveal the formation of stable interactions between (S)-crizotinib and specific residues within the MTH1 active site (e.g., Tyr7, Phe27, Phe72, Trp117). These interactions contribute to the high specificity and potency of (S)-crizotinib to inhibit MTH1 [8,12].

Molecular dynamics (MD) simulations further support the superior binding properties of (S)-crizotinib. These simulations demonstrate greater stability and lower fluctuations in the (S)-crizotinib/MTH1 complex compared to the complex formed between (R)-crizotinib and MTH1. Additionally, binding free energy calculations solidify the notion of higher binding affinity and stability of (S)-crizotinib within the MTH1 active site [12].

### 2.5. (S)-Crizotinib Induces DNA Damage Response and ER Stress

(S)-crizotinib treatment inhibits MTH1, leading to an elevation of DNA single-strand breaks and activation of DNA damage response (DDR) pathways. Key DDR proteins, such as ATM and ATR, along with their downstream effectors, like p53, become activated. This activation cascade leads to cell cycle arrest and apoptosis, ultimately promoting cell death [8].

### 2.6. (S)-Crizotinib Induces Apoptosis via the ER Stress Pathway

(S)-crizotinib induction of an increase in ROS levels also triggers the activation of the ER stress pathway. The pathway involves the phosphorylation of eIF2α, which subsequently leads to the activation of ATF4 and CHOP, which are both critical factors in ER stress-mediated apoptosis. The activation of this pathway further contributes to the pro-apoptotic effects of (S)-crizotinib in cancer cells [9].

### 2.7. In Vitro and In Vivo Studies Support the Therapeutic Potential of (S)-Crizotinib

(S)-crizotinib significantly reduces cell viability and colony formation across various cancer cell lines, including those with KRAS mutations. Notably, it demonstrates efficacy in gastric cancer (GC), independent of MTH1 inhibition [13]. This MTH1-independent mechanism appears to involve the induction of oxidative DNA damage and apoptosis, as evidenced by increased markers like γH2AX and p53BP1 foci [13]. Furthermore, (S)-crizotinib markedly suppresses tumor growth in SW480 colon carcinoma xenograft models with minimal observed toxicity in non-cancerous tissues, suggesting a favorable safety profile [8,9,10]. These findings collectively highlight the promise of (S)-crizotinib as a therapeutic strategy for MTH1-dependent tumors and potentially for MTH1-independent cancers, like gastric cancer.

**Figure 1 cancers-16-02479-f001:**
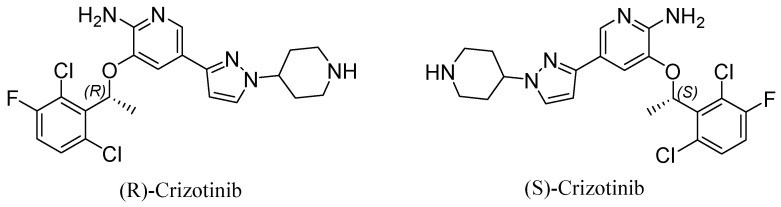
Chemical structure of (R)-crizotinib (**left**), and (S)-crizotinib (**right**). While (S)-crizotinib demonstrated initial promise, (R)-crizotinib emerged as the clinically preferred enantiomer due to its distinct properties. This review will therefore concentrate on the functions of (R)-crizotinib. Thus, crizotinib will refer to (R)-crizotinib in this context.

## 3. Crizotinib as Targeted Therapy in Cancer

Cancer treatment is undergoing a significant transformation with the emergence of targeted therapy. This approach represents a paradigm shift from traditional chemotherapy as it focuses on specific molecular vulnerabilities within cancer cells. In contrast to the broad-spectrum effects of chemotherapy, targeted therapies offer the potential for greater efficacy and a more favorable side effect profile. By leveraging advancements in our understanding of the genetic mutations that underlie cancer, researchers are developing drugs that precisely target these abnormalities. This personalized approach holds great promise for improved treatment outcomes.

Crizotinib is used as a targeted agent and serves as a first-generation multi-target kinase inhibitor that is clinically approved as an ALK (anaplastic lymphoma kinase) and ROS1 (c-ros oncogene 1) inhibitor [14]. Crizotinib also inhibits the c-Met/hepatocyte growth factor receptor [3]. Crizotinib is FDA-approved for the treatment of patients with locally advanced or metastatic non-small cell lung cancer (NSCLC) that is ALK-positive [1] or ROS1-positive [2] and for adult and pediatric patients one year of age and older with incurable, recurrent, or refractory inflammatory (ALK)-positive myofibroblastic tumors (IMTs) [4] (Figure 2). Crizotinib has been shown to inhibit the activity of Janus kinase 2 (JAK2), a non-receptor tyrosine kinase [15], in addition to native and T315I-mutated Bcr/Abl [16] (Figure 2).

Crizotinib also demonstrated potent activity against c-Met kinase, which is involved in carcinogenesis [17]. The c-Met proto-oncogene was first identified in osteosarcoma cells exposed to carcinogens [18]. High levels of c-Met amplification and the activation of Met mutations or fusions have now been shown to be oncogenic drivers [17,19]. Although Met is expressed in numerous normal cells, it is overexpressed in many human malignancies, such as gastrointestinal, lung, and breast cancers [20]. Met plays a role in the development and progression of certain human malignancies and mediates proliferation, migration, and invasion [18]. C-Met has, therefore, been successfully used as a biomarker for diagnosis and prognosis, survival, postoperative recurrence, risk assessment, and pathologic grading, as well as a therapeutic target. In addition, recent work suggests that the inhibition of Met expression and function has potential clinical benefits [18] (Figure 2).

Recently, crizotinib has also been proposed to treat triple-negative breast cancer (TNBC), an aggressive breast cancer subtype lacking effective targeted therapies. Indeed, c-Met [21] and ROS1 [22] expression levels were found to be higher in TNBC tissues compared to normal tissues. Thus, targeting these biomarkers with crizotinib has shown promise in preclinical studies, laying the foundations for clinical trials (ClinicalTrials.gov Identifier: NCT03620643). A crosstalk between c-Met and EGFR has been implicated in therapeutic resistance. Dual inhibition of c-Met and EGFR has been proposed as a therapeutic strategy [23]. A synergistic anti-tumor effect was observed using a FAK inhibitor and crizotinib in TNBC xenograft models and human TNBC organoid models, characterized by an upregulation of p-FAK [22].

## 4. Crizotinib as an Allosteric Kinase Inhibitor

There are various strategies for designing enzyme inhibitors, each targeting a distinct aspect of the enzyme’s function. Competitive inhibitors, which are designed to mimic the natural substrate ATP, bind directly to the ATP-binding pocket. This effectively blocks access to the substrate and stops enzymatic activity. Allosteric inhibitors, on the other hand, employ an indirect approach. By binding to a distinct regulatory site, they induce conformational changes that disrupt the active site without occupying it directly. This targeted difference can lead to increased selectivity for allosteric inhibitors. The ATP-binding pocket is conserved in many enzymes; therefore, it is a common target. However, allosteric sites often have greater structural diversity and thus provide a more specific binding niche for drug development (Figure 3).

Reports of the crizotinib’s ability to inhibit T315I-mutated Bcr/Abl suggest that this activity [16] is attributed to its ability to interact allosterically with Bcr/Abl (Figure 2 and Figure 3). Therefore, it is possible that crizotinib has two binding affinities: one to the ATP-binding pocket and another for allosteric binding to other functionally important sites of the respective kinases. Although it has been demonstrated that crizotinib can overcome the gatekeeper mutation of Bcr/Abl [16], it is reasonable to assume that crizotinib can also affect other gatekeeper mutations in other target kinases.

## 5. Crizotinib Synergizes with Cancer Therapy

Crizotinib has shown promise in overcoming chemoresistance when used in combination therapy. Research has highlighted its efficacy in various cancers. Studies have shown that the combination of crizotinib with other chemotherapeutic agents can restore sensitivity to treatment, particularly in cases of acquired resistance. For example, Krytska et al. (2015) demonstrated that the combination of crizotinib with topotecan and cyclophosphamide improved tumor responses in neuroblastoma models [24]. Greengard et al. (2015) supported this finding and showed that crizotinib in combination with conventional chemotherapy was safe and tolerable in children with refractory solid tumors and anaplastic large cell lymphoma [25].

Moreover, it has been recently reported that crizotinib nanomicelles effectively enhanced doxorubicin-elicited anticancer efficacy in a p53Y220C pancreatic cancer via the degradation of the p53 mutant, which was induced by crizotinib [26]. These results suggest that the use of crizotinib in combination therapies could be a valuable strategy to overcome chemoresistance in various malignancies and open new avenues for effective treatment approaches.

It is known that high c-Met expression is associated with a poor prognosis in ovarian cancer [27]. C-Met activation has been associated with a poor prognosis in patients with high-grade serous ovarian cancer (HGSOC) [28]. Crizotinib, in synergy with platinum compounds, inhibits the growth of ovarian cancer cells in vitro and in vivo [29]. In addition, crizotinib synergizes with carboplatin to enhance apoptosis in ovarian cancer [30]. Consistent with preclinical research, crizotinib demonstrated effective activity in patients with ovarian cancer with a ROS1 rearrangement in a case study [31].

Crizotinib is thought to exert its effect on cancer cells by modulating the growth, migration, and invasion of malignant cells [32]. Other studies suggest that crizotinib may also act by inhibiting angiogenesis in malignant tumors [33,34,35,36].

## 6. Crizotinib Inhibits TGFβ Signaling Pathway

The transforming growth factor-beta (TGF-β) pathway, crucial for cellular processes, like proliferation and migration, is initiated by a ligand binding to a receptor complex. This activates Smad proteins, which then translocate to the nucleus and regulate gene transcription [37,38,39]. Emerging evidence suggests a crosstalk between the TGF-β signaling pathway and the c-Met signaling pathway (Figure 2). Activation of c-Met can influence TGF-β signaling [40]. Conversely, targeting c-Met with inhibitors like crizotinib may affect the TGF-β signaling pathway. This modulation holds therapeutic potential in cancers where TGF-β promotes tumor progression [41]. Crizotinib treatment could potentially lead to reduced Smad-mediated gene transcription, thereby inhibiting tumor cell invasion and metastasis, as supported by a recent study by Park et al. (2022), demonstrating its ability to suppress TGF-β signaling in NSCLC cells [41].

The crosstalk between c-Met and TGF-β signaling holds particular significance for comprehending the multifaceted effects of crizotinib in cancer treatment, particularly in tumors exhibiting dysregulation of both pathways (Figure 2). For example, in hepatocellular carcinoma (HCC), the c-Met and TGF-β pathways collaborate to drive a epithelial-mesenchymal transition (EMT), a crucial step in cancer metastasis that enables tumor cells to acquire invasive properties [42,43].

## 7. Crizotinib Overcomes Tumor Microenvironment-Mediated Drug Resistance

Cancer therapy faces a major challenge in the form of chemoresistance. This phenomenon occurs when cancer cells develop the ability to not only survive but possibly even proliferate in the presence of chemotherapeutic agents [44]. Chemoresistance is a complex issue with various underlying mechanisms that significantly impair the efficacy of cancer treatment [45].

Two main categories of chemoresistance are recognized: intrinsic resistance and resistance mediated by the tumor microenvironment.

Intrinsic chemoresistance: This type of resistance arises from changes within cancer cells. These changes may include genetic mutations in genes related to drug uptake, metabolism, or cell death pathways [46]. Basically, cancer cells make intrinsic changes to their internal systems and molecular mechanisms that allow them to evade the cytotoxic effects of chemotherapeutic agents [47].

Tumor microenvironment-mediated chemoresistance: In contrast to intrinsic resistance, this type of resistance does not involve changes within cancer cells. Instead, the tumor microenvironment, a complex ecosystem of stromal cells, immune cells, and signaling molecules surrounding the tumor, plays a critical role [48]. Elements within this microenvironment can create a protective niche for cancer cells. This can manifest itself through mechanisms such as impeding drug delivery, promoting pro-survival signals, or even facilitating the emergence of pre-existing or treatment-resistant subpopulations within the tumor [49].

Understanding the intricate mechanisms underlying chemoresistance is critical to improving treatment outcomes. This knowledge can pave the way for the development of novel therapeutic strategies that can overcome these resistance mechanisms and ultimately improve the fight against cancer.

Crizotinib, unlike imatinib, overcomes soluble factor-mediated drug resistance in chronic myelogenous leukemia (CML) cell lines [15]. The ability of crizotinib to overcome soluble factor-mediated CML drug resistance is attributed to its ability to inhibit JAK2. JAK2 is a kinase that plays a significant role in mediating tumor microenvironment (TME) drug resistance, which may be important for prevailing residual disease in CML and leukemia (Figure 2).

The ovarian cancer (OC) TME, such as malignant ascites, has been associated with drug chemoresistance [50]. Recent findings indicate that exposure to OC ascites led to platinum chemoresistance in OC cells in 11 out of 13 cases (85%) during an ex vivo experiment [6].

In contrast, 75% of cirrhotic ascites did not lead to platinum chemoresistance in OC cells. Cytokine array analysis of malignant ascites showed that IL-6 and, to a lesser extent, the hepatocyte growth factor (HGF) were enriched in OC ascites. Crizotinib, as an inhibitor of the HGF/c-Met and IL-6/JAK signaling pathways, was effective in restoring the platinum chemosensitivity of OC [6] (Figure 2). Overall, these studies suggest that crizotinib, in combination with other therapies, can be effective in overcoming chemotherapy resistance in cancer.

## 8. Crizotinib and Multidrug Resistance in Cancer

Multidrug resistance (MDR) presents a formidable challenge in oncology. It signifies the ability of cancer cells to develop resistance to a broad spectrum of chemotherapeutic agents, rendering them ineffective and hindering treatment efficacy. This phenomenon is a major contributor to treatment failure, tumor recurrence, and poorer patient prognosis [51]. The mechanisms underlying MDR are multifaceted, encompassing the enhanced efflux of drugs from the cell, increased DNA repair capacity, and the evasion of cell death pathways [52]. P-glycoprotein (P-gp) acts as a key mediator of MDR [53]. This transmembrane efflux pump actively expels chemotherapeutic agents from cancer cells, thereby reducing their intracellular concentration and compromising their cytotoxic effects [54]. Notably, P-gp’s broad substrate specificity allows it to efflux a diverse range of drugs, contributing to the “multidrug” nature of MDR [54]. Elucidating the mechanisms by which P-gp fuels MDR is paramount for developing novel therapeutic strategies in cancer treatment. These strategies may encompass the inhibition of P-gp activity to enable drug accumulation within cancer cells, or the design of new chemotherapeutics less susceptible to P-gp-mediated efflux [55]. By overcoming P-gp-mediated MDR, researchers aim to improve the efficacy of chemotherapy and ultimately enhance patient prognosis.

Zhou et al. (2012) reported that crizotinib could reverse the MDR of cancer cells by inhibiting the function of P-glycoprotein [56] (Figure 2). This study investigated the effect of crizotinib on MDR mediated by ATP-binding cassette subfamily B member 1 (ABCB1), also known as P-glycoprotein. The findings revealed that crizotinib reverses MDR by inhibiting ABCB1 transport function, without altering ABCB1 expression levels or affecting the Akt and ERK1/2 signaling pathways. These results suggest a promising role for crizotinib in combination chemotherapy regimens, potentially enhancing the efficacy of conventional chemotherapeutic drugs for MDR-positive cancers.

## 9. Synergy of Crizotinib with PARP Inhibitors

Poly (ADP-ribose) polymerase (PARP) is crucial for DNA repair, particularly for the base excision repair (BER) pathway. When BER is compromised, inhibiting PARP can lead to the accumulation of single-strand breaks (SSBs) that convert to complex double-strand breaks (DSBs) [57]. Patients with BRCA gene mutations are particularly susceptible to PARP inhibitor therapy, as it disrupts both BER and homologous recombination (HR), causing DNA damage [58]. PARP inhibitors target BRCA-mutated cancer cells and enhance the cytotoxicity of chemotherapy drugs that induce SSBs. They have shown efficacy as single agents in BRCA1/2-associated ovarian and breast cancers and potentially when combined with chemotherapy in treating triple-negative breast cancer [59]. All three FDA-approved PARP inhibitors (Olaparib, Niraparib, and Rucaparib) show potential synergy with crizotinib by preventing DNA repair in cancer cells already stressed by crizotinib-induced damage.

Crizotinib was reported to synergize with PARP inhibitors (PARPi) for treating OC-bearing mutations in BRCA genes. C-Met is associated with PARP1 and phosphorylates it at Tyr907 (PARP1 pTyr907 or pY907). PARP1 pY907 increases the enzymatic activity of PARP1 and decreases its binding ability to a PARP inhibitor, making cancer cells resistant to PARP inhibition. Indeed, c-Met-mediated PARP phosphorylation was recently shown to confer PARPi resistance in preclinical breast cancer models [60]. The combination of c-Met and PARP1 inhibitors synergistically suppressed breast cancer cell growth in vitro and in xenograft tumor models. Similar synergistic effects were reported in a lung cancer xenograft tumor model [60]. Additionally, crizotinib enhanced the efficacy of PARP inhibitors in OC cells and xenograft models by inducing autophagy [61]. These results suggest a potential synergy between PARP inhibitors and crizotinib (Figure 2). This combination could be considered in patients to limit resistance to PARP inhibitors, as shown in the latest studies [62].

In addition, experimental outcomes showed that the combination of crizotinib and PARP inhibitors resulted in the activation of the ATM/CHK2 pathway and in the inhibition of the c-Met pathway, which contributed to a decrease in RAD51 levels and the induction of caspase-3-dependent apoptotic cell death. These findings suggest that the combination of crizotinib and a PARP inhibitor could be considered and further explored as a novel therapeutic strategy in HGSOC [30].

## 10. Crizotinib and Cancer Stem Cells (CSCs)

Cancer Stem Cells (CSCs) play a crucial role in chemotherapy resistance and contribute to tumor relapse and aggressive cancer progression. CSCs exhibit properties such as quiescence, enhanced DNA repair mechanisms, and altered cell cycle checkpoints, which enable them to survive conventional chemotherapies that primarily target rapidly dividing cells [63,64]. In addition, the CSC niche—consisting of immune cells, mesenchymal stem cells, endothelial cells, and cancer-associated fibroblasts—significantly supports the maintenance of CSC properties and increases chemotherapy resistance. Interactions between CSCs and the tumor microenvironment, enriched with growth factors and cytokines, further enhance CSC chemoresistance by activating survival pathways, DNA repair mechanisms, and anti-apoptotic proteins, ultimately leading to treatment failure and disease progression [63,64].

Pretreatment with (R)-crizotinib has shown promising results in suppressing the proliferation of CSCs in vitro and inhibiting tumor growth in vivo. Administration of (R)-crizotinib in vivo resulted in a 72% reduction in the growth of tumors derived from CSC-like cells [65]. The morphologic changes induced by (R)-crizotinib were accompanied by significant alterations in the expression of CSC-specific marker proteins such as CD44, ALDH1, Nanog, and Oct-4A [65]. These changes indicate that the cellular properties of CSCs are significantly affected by (R)-crizotinib [65] (Figure 2).

In addition, (R)-crizotinib significantly altered the expression levels of Snail, Slug, and E-cadherin, which were considered markers of CSCs. Analysis of several signal transduction molecules indicated that AMPK phosphorylation appeared to be selectively inhibited by (R)-crizotinib. This suggests that R-crizotinib may reduce the properties of CSCs by inhibiting AMPK activity [66] (Figure 2). Interestingly, some natural products might mimic the activity of crizotinib in inhibiting CSCs [67].

## 11. Crizotinib Degrades Missense Mutant p53

Missense mutant p53 (mut-p53) is stabilized in most cancer cells. Cancer cells harboring mut-p53 are dependent on mut-p53 for survival and proliferation. Preclinical studies have shown that knocking out the p53 mutation prolongs the survival of mice and suppresses tumor growth [68]. This suggests that knocking out the p53 mutation is a promising therapeutic approach. However, the genomic depletion of mut-p53 is not yet available for cancer therapy in the clinic. Most studies focus on pharmacological approaches to induce the degradation of the p53 mutation via the proteasome or the lysosome [69]. An example of a group of agents that utilize the proteasome to induce p53 mutation degradation are Hsp90 inhibitors, such as geldanamycin [70] and 17AAG [71].

In a recent preclinical study, crizotinib exhibited the ability to degrade mutant p53 protein (mut-p53) by inducing its ubiquitin-dependent proteasomal degradation. This effect is observed across a wide range of mut-p53 variants [26] (Figure 2). This finding is significant for cancer therapy in general. As a tumor suppressive factor, p53 is activated to induce DNA repair, cell cycle arrest, or apoptosis when cells are exposed to DNA-damaging agents such as platinum compounds. Somatic TP53 mutations occur in practically every kind of cancer, their frequencies range from 38 to 50% in ovarian, esophageal, colorectal, head and neck, throat, and lung cancers to approximately 5% in primary leukemia, sarcoma, testicular cancer, malignant melanoma, and cervical cancer [72,73,74]. Interestingly, about 96% of HGSOC harbors somatic p53 mutations. Most p53 mutations found in OC are missense mutations [75].

The mutated p53 is stabilized in the cancer cells and is difficult to degrade. This mutated p53 protein not only loses the function of its wild-type form to prevent tumorigenesis (loss of function, LOF), but also acquires tumor-promoting activities (gain of function, GOF), such as an increased ability for proliferation, migration, and drug resistance [75]. Since attempting to induce the degradation of mut-p53 is an attractive strategy to realize in precision anti-tumor therapy, there is currently no FDA-approved drug targeting mut-p53 in cancer. Treatment of cancer cells harboring mutant p53 (mut-p53) with crizotinib can trigger ubiquitin–proteasome pathway-mediated degradation of mut-p53, thereby abrogating its GOF properties. This treatment results in decreased cell proliferation, migration, death, and cell cycle arrest, and increased sensitivity to activity of DNA-damaging agents in cancer with mut-p53 [26].

Pancreatic cancer, one of the deadliest cancers, also contains considerably high levels of mutant p53 [76]. Crizotinib showed a chemo-sensitizing effect on p53-mutated cancer in a BxPC-3 pancreatic cancer model [26]. Taken together, the above results illustrated the feasible and promising avenue for the treatment of cancers harboring mutant p53 such as HGSOC and others [77,78,79,80]. The capacity of crizotinib to indirectly induce the degradation of the p53 mutant, as demonstrated in preclinical studies, has yet to be confirmed in clinical settings.

In patients with ALK-rearranged NSCLC, co-occurring TP53 mutations are predictive of an unfavorable outcome to systemic therapy [81]. These mutations negatively impact the response to crizotinib and are associated with shorter progression-free survival (PFS) in patients with ALK-rearranged NSCLC [82]. It is of significant interest to further investigate the interaction between ALK activation and p53 mutation in clinical settings, and to assess their response to crizotinib or other ALK inhibitors. This analysis could help clarify the observed discrepancies between preclinical and clinical observations.

To overcome mut-p53 induced drug resistance, Bi et al. have recently proposed to encapsulate Fluplatin in nanoparticles (FP NPs). They reported a significant therapeutic effect of FP NPs on p53-mutant NSCLC, which was exerted by degrading mut-p53 [83]. Also, TNBCs bearing stable mutant p53 are dependent on mutant p53 for growth [84] and elevated levels of c-Met. Stabilized mutant p53 cells correlate with highly proliferative human TNBCs of poor outcome [85]. Thus, a combination of therapeutic strategies leading to the degradation of mut-p53 should be further investigated [86].

## 12. Crizotinib Exhibited Activity That Is beyond Oncology

Similarly to the threat of drug resistance in cancer cells, drug resistance in bacteria is also becoming a serious threat to public health worldwide, and the discovery of novel antibacterial compounds is urgently needed. Moreover, cancer drug resistance and bacterial drug resistance share several mechanistic similarities. Both types of resistance arise through evolutionary pressures that select for resistant cells or microorganisms capable of surviving therapeutic interventions. Furthermore, genetic mutations, epigenetic modifications, and the activation of alternative signaling pathways that bypass the effects of treatment are factors involved in conferring drug resistance in cancers and bacteria [43].

Crizotinib, an established anticancer drug, exhibits unexpected antibacterial activity against Gram-positive bacteria. Crizotinib exhibited a unique mechanism of action that involved disrupting bacterial cellular metabolism by inhibiting ATP production, which is essential for various bacterial functions. Thus, crizotinib compromises survival and proliferation of bacteria. In addition, crizotinib targets CTP synthase, a key enzyme in pyrimidine metabolism. Pyrimidine nucleotides are crucial for bacterial DNA synthesis. Crizotinib’s inhibition of CTP synthase disrupts pyrimidine biosynthesis, hindering the bacteria’s ability to produce the DNA required for replication [87]. Moreover, crizotinib can effectively reduce the development of drug resistance in Staphylococcus aureus. Hence, crizotinib can be a promising option for combating multidrug-resistant bacterial infections [87]. Furthermore, the antibacterial effects of crizotinib have been demonstrated in animal models, where it increased the survival rate of infected mice and decreased pulmonary inflammation activity, highlighting its potential for treating bacterial infections [87]. Crizotinib’s antibacterial activity against Gram-positive bacteria presents a novel approach to combatting infectious diseases. This discovery opens doors for innovative therapeutic strategies and a potential paradigm shift in antibacterial drug development.

## 13. Off-Target Toxicities of Crizotinib

Crizotinib offers significant benefits in cancer therapy but is associated with notable off-target toxicities. The most common adverse events reported following crizotinib administration are vision disorder (occurring in 71% of the patients), diarrhea (in 61%), and edema (in 49%) [88]. Other gastrointestinal side effects are less common and include stomatitis, nausea, vomiting, constipation, and mild liver dysfunction [88]. Liver function monitoring is crucial as crizotinib can elevate liver enzyme levels [89]. Hematologic adverse events related to crizotinib treatment that are associated with bone marrow include anemia (in 32%) and neutropenia (in 30%). Most adverse events are grade 1 or 2 in severity [88]. Grade 3 or 4 elevations of aminotransferase levels occur in (14%) and of neutropenia occur in 11% of patients [88]. Interstitial lung disease is a serious side effect, occurring in 1% of patients and resulting in the permanent discontinuation of crizotinib treatment [88].

Beyond these documented effects, high-dose crizotinib therapy has been shown to trigger immunogenic cell death (ICD) in cancer cells lacking the intended ALK/ROS1 mutations, suggesting potential off-target effects beyond the drug’s primary mechanism [90,91]. Additionally, a case study reported fulminant hepatotoxicity in a young patient receiving a supratherapeutic dose of crizotinib, highlighting the possibility of idiosyncratic, unpredictable reactions [92]. Finally, the importance of understanding drug interactions is underscored by reports of cardiac toxicity arising from the combination of crizotinib with sofosbuvir/velpatasvir [93]. This pharmacokinetic interaction emphasizes the need for the careful management of concomitant medications to minimize adverse effects in patients receiving crizotinib in combination with other drugs.

## 14. Conclusions

Crizotinib, a tyrosine kinase inhibitor, shows efficacy in the treatment of ALK-positive or ROS1-positive non-small cell lung cancer (NSCLC) and refractory inflammatory myofibroblastic tumors (IMTs) with ALK mutations.

Crizotinib is available in two enantiomers: (R)-crizotinib, the primary therapeutic form, and (S)-crizotinib, which has intriguing activity against the DNA repair enzyme MTH1.

Crizotinib’s mechanism of action is based on the competitive and potentially allosteric inhibition of ATP binding in kinases such as ALK, ROS1, c-Met, JAK2, and Abl. In addition to its established role in approved cancer therapy, this review highlights crizotinib’s ability to overcome cancer cell chemoresistance by interfering with P-glycoprotein and PARP enzymes. Moreover, crizotinib also exhibits antibacterial activity against drug-resistant strains.

Crizotinib shows strong synergy in combination with other chemotherapies, especially in refractory tumor cells. Crizotinib shows synergistic effects with PARP inhibitors in BRCA-mutated ovarian cancer. This drug can target critical vulnerability in many p53-mutated cancers. The p53 mutant, in contrast to its wild-type form, promotes cancer cell survival. Crizotinib can induce the degradation of the p53 mutant, sensitizing these cells to DNA-damaging agents and triggering apoptosis.

Overall, crizotinib exerts multiple anti-tumor effects by inhibiting kinases and restoring drug sensitivity. This underscores its potential for combination therapies in cancers with a high prevalence of the p53 mutant, such as triple-negative breast cancer and high-grade serous ovarian cancer.

## Figures and Tables

**Figure 2 cancers-16-02479-f002:**
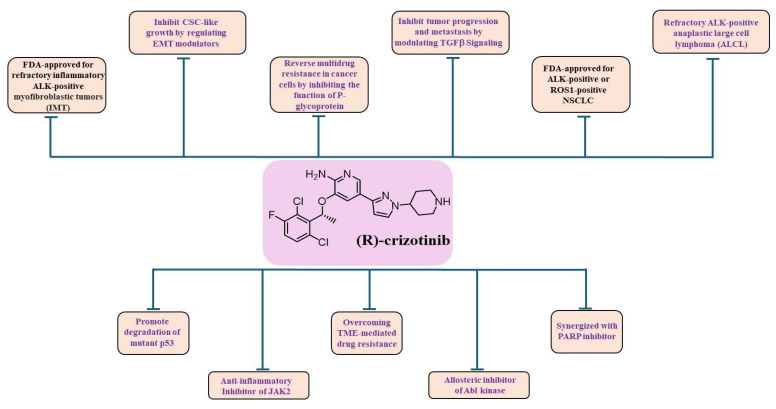
Crizotinib activity. (R)-crizotinib structure is shown. FDA-approved indications are present in black writing. The abilities to modulate activity of other enzymes and processes are present in purple writing.

**Figure 3 cancers-16-02479-f003:**
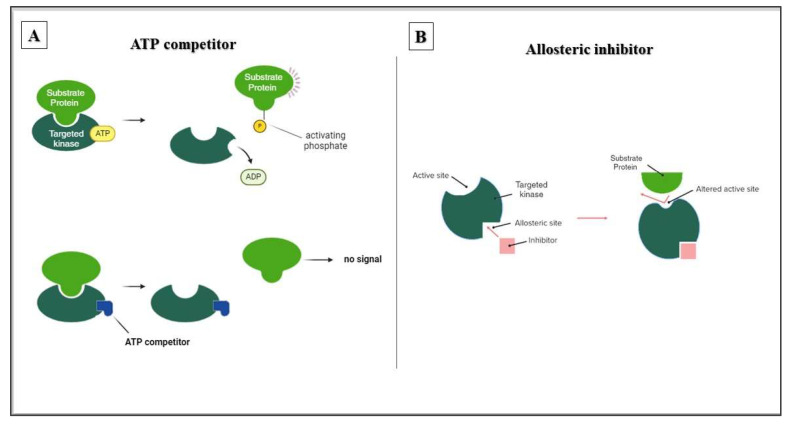
Comparison of ATP competitive and allosteric inhibitors of kinases. This figure illustrates the distinct mechanisms by which ATP competitive and allosteric inhibitors regulate kinase activity. (**A**) The active site of the kinase is depicted, highlighting the ATP binding pocket crucial for phosphorylating target proteins. ATP competitive drugs directly compete with ATP for binding at this site, thereby inhibiting kinase activity. (**B**) The action of allosteric inhibitors. These inhibitors bind to a different regulatory site on the enzyme. Upon binding, they induce conformational changes that indirectly affect the ATP binding pocket, hindering ATP binding and ultimately inhibiting enzymatic activity.

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
