# Peer review of "Overcoming Chemoresistance in Cancer: The Promise of Crizotinib"

_cancers, 2024, doi:10.3390/cancers16132479_

Round 1

Reviewer 1 Report

Comments and Suggestions for Authors

This review describes the multiple anti-tumor effects of crizotinib, underscoring its potential for combination therapies. Additionally, it emphasizes the distinct mechanisms of action of the two enantiomeric forms of the drug. Although the topic is very interesting, a more comprehensive discussion of the pleiotropic effects of crizotinib should be warranted, including an exploration of all documented and potential off-target toxic effects in humans.

Specific Comments

Line 43 and Figure 2: Please include that crizotinib is also approved for the treatment of relapsed/refractory ALK-positive anaplastic large cell lymphoma (ALCL). 

Figure 2: Please correct the typos to read as "TGFbeta" instead of "RGFbeta".

Line 169: Please correct the sentence. 

Lines 233-248: Please correct the sentence. Explain how c-MET modulates SMAD phosphorylation, including relevant citations. Avoid repetitions. Additionally, describe that crizotinib could also suppress SMAD phosphorylation in an ALK/MET/RON/ROS1-independent manner (PMID: 35999455).

Line 372 and 382: Please avoid implying that crizotinib actively degrades mutant p53. As correctly stated, crizotinib mediates its degradation through the ubiquitin-proteasome pathway.

Lines 407-410: It is stated that TP53 mutation in ALK-rearranged NSCLC negatively affects the response to crizotinib. Based on the hypothesis that crizotinib could enhance p53 degradation, one would expect the opposite effect. Please clarify this apparent contradiction. 

Lines 417-437: The discussion on natural compounds targeting mutant p53 is unrelated to crizotinib and the scope of the present review. Please remove this section.

Author Response

My response is attached.

Reviewer 2 Report

Comments and Suggestions for Authors

This review focused on the mechanisms underlying the potential of crizotinib to overcome chemoresistance in cancers. The information presented was not convincing that crizotinib could be used to overcome chemoresistance though. 

In Section 2, "Crizotinib Stereoisomers", the authors reviews the antitumor activity and associated mechanisms for (S)-crizotinib only even though studies reviewed in other sections used racemic crizotinib. If comprehensive reviews of (R)-crizotinib have been published elsewhere, there should be a short paragraph to give a brief summary and provide references. 

Line 168: Please provide the reference to "Inh strategies".

Line 218-219. "Other studies suggest that ...", which imply that there should be more than one study, right? Please provide more references. 

Line 240-247. Reference #37 and #38 did not involve crizotinib. Please cite the references related to the drug of interest. Otherwise, the entire paragraph is just the authors' speculation without evidence. 

Line 358-364. Please provide the references. 

Comments on the Quality of English Language

Minor editing of English language required.

Author Response

My response is attached

Round 2

Reviewer 1 Report

Comments and Suggestions for Authors

The authors effectively addressed all queries raised and improved their manuscript.

Comments on the Quality of English Language

The authors effectively addressed all queries raised and improved their manuscript.

Reviewer 2 Report

Comments and Suggestions for Authors

The authors have addressed all my concerns.